# Analysis of Early Growth of Piglets from Hyperprolific Sows Using Random Regression Coefficient

**DOI:** 10.3390/ani13182888

**Published:** 2023-09-11

**Authors:** Dubravko Škorput, Nina Jančo, Danijel Karolyi, Ana Kaić, Zoran Luković

**Affiliations:** 1Divison of Animal Science, Faculty of Agriculture, University of Zagreb, Svetošimunska cesta 25, 10000 Zagreb, Croatia; dkarolyi@agr.hr (D.K.); akaic@agr.hr (A.K.); lukovic@agr.hr (Z.L.); 2Family Enterprise Jančo, Matije Gupca 19, 31424 Punitovci, Croatia

**Keywords:** hyperprolific sows, birth weight, growth, random regression coefficient

## Abstract

**Simple Summary:**

The increase in litter sizes in pigs obtained through selection has introduced challenges to breeders such as birth weight variability, piglet nursing, and piglet survival. Differences in birth weights can affect the later growth of piglets. We analysed the most important effects on the growth of piglets born in hyperprolific herds using a random regression coefficient model, which provides a smoother estimation of parameters compared to traditional fixed effects models and accounts for heterogeneous variances between measurements. Birth weight was the most influential factor on the final weight at 85 days of age. Litter size and parity also showed significant effect on the final weight. The results obtained using a random regression coefficient model could be encouraging for further application in pig growth analysis due to the ability of the model to describe individual growth patterns of piglets of variable birth weights and estimate the future growth of such piglets. In addition, the practical contribution of the paper is deeper insight into growth patterns of piglets from highly prolific sows under farm conditions, focusing on the need to control the variability of birth weight of large litters.

**Abstract:**

Management of hyperprolific sows is challenging when it comes to controlling birth weight variability and piglet survival in large litters. The growth of low birth weight piglets can be compromised and have a negative impact on production efficiency. The objective of the study was to apply a random regression coefficient model to estimate the main effects of the growth of piglets of highly prolific sows. The dataset contained growth data for 360 piglets from 25 Pen Ar Lan Naima sows. In addition to routine procedures after farrowing, piglets were weighed five times: on day 1 after farrowing, on day 14 of life, at weaning on day 28, on day 30 of nursery period, and at the end of the nursery period when piglets were 83 days old. Data were treated as longitudinal, with body weight as the dependent variable. Fitting age as a quadratic regression within piglets in the random part of the model helped to determine the significant effect of birth weight, litter size, and parity on the growth of the piglets. Since the piglets from large litters often have non-uniform birth weights and this can affect further growth, the use of a random regression coefficient model is practical for analysing the growth of such piglets due to the ability to describe the individual growth pattern of every individual.

## 1. Introduction

In recent decades, selection for improved litter size in pigs has led to a significant increase in the total number of piglets born and the number of piglets born alive, mostly from hyperprolific sow lines [1]. However, in addition to positive effects of increased litter size, the development of highly prolific hybrid lines resulted in changes that required the adaptation of existing approaches in farm management, with a focus on nursing piglets of low viability, usually those with low birth weights [2]. The variability of birth weight and the lower survival rate of piglets with low birth weights are now recognised as one of the key problems in the management of hyperprolific sows [3,4]. In addition, common beliefs regarding the high productivity of hyperprolific sows can also be questioned in terms of the number of teats and longer farrowing durations [5].

Piglet growth is influenced by litter size while the piglets are still in the uterus: larger litters result in intrauterine crowding and embryos that are implanted first may physically prevent the development of additional embryos [5] resulting in a higher number of light piglets. Impaired intrauterine growth is strongly correlated with litter size [6]. Since growth rate strongly determines the economic efficiency of pig production, it is important to assess numerous factors that influence this trait. Since growth is a longitudinal trait that can be measured several times during the lifespan, a traditional approach with single trait linear models may not be sufficient to describe the nature of the trait. The reason for this is that the variability between measurements on the same experimental unit might change through time and the measurements may therefore be characterised by heterogeneous variances [7]. Because of their properties, random regression models can be used to analyse such traits. The main advantages of random regression models over conventional models are the improved accuracy of estimation, the avoidance of adjustment of phenotypic data, a smaller number of parameters to describe longitudinal measurements, smoother (co)variance estimates, and the possibility to estimate covariance components and to predict breeding values at any point along the trajectory [8,9,10].

Random regression models have typically been applied to pigs in estimating breeding values for growth [11], feed intake [12], and fertility traits [13]; however, the application of a random regression coefficient might also be useful in the estimation of non-genetic factors [14] for traits with a longitudinal structure.

The aim of the study was to examine the possibility of using a random regression coefficient model to determine the main effects on body weight of piglets from hyperprolific sows during the growth period from birth to 85 days of age.

## 2. Materials and Methods

### 2.1. Experimental Design and Data

The animal study protocol was approved by the Ethics Committee of the University of Zagreb, Faculty of Agriculture. All animals originated from commercial production facilities. No measurements were made that were outside of the standard industry animal husbandry techniques, and the animals were cared for in compliance with local and EU legal standards. The health and welfare of all animals were monitored throughout the sampling days by farm staff, according to the farms’ standard operating protocols and veterinary recommendations. The experiment was conducted on a family farm with hyperprolific sows. The experimental herd consisted of 25 Pen Ar Lan Naima sows. Gestating sows were moved to the farrowing rooms a week before the expected farrowing day. Sows were treated with d-cloprostenol on day 112 of the gestation period [15]. The use of farrowing induction on the farm in question is a routine procedure that has an important function in effectively monitoring farrowing. Each pen was equipped with a commercial farrowing crate. An infrared heat lamp was placed in each farrowing pen to provide additional heat to piglets. On average, the piglets were weaned at 28 days of age. From day 5 after farrowing until weaning, the sows were fed ad libitum. Within the first 18 h of life, the piglets were individually weighed. The litter size was balanced by cross-fostering within two days after farrowing by transferring piglets with lower birth weights. Male piglets were castrated on the third day of life. During the experiment, 360 piglets (Naima sows × P76 Pen Ar Lan hybrid boars) from 25 litters were weighed five times: on the 1st day after farrowing (BW), on the 14th day of life (W2), at weaning on the 28th day (W3), on the 30th day of nursery period (W4), and at the end of the nursery period when the piglets were 83 days old (W5). After farrowing, the sows were fed with the standard feed mixture for lactating sows. Piglets were fed standard feed mixtures: pre-starter, starter, grower. The detailed chemical composition of the mixtures can be found in Table 1.

The litter size was presented as the number of piglets born alive (NBA) and categorised into five groups. The distributions of the litter size groups are given in Table 2. Also, for analysis purposes, BW was categorised into five classes: ≤1000 g, 1001–1200 g, 1201–1400 g, 1401–1600 g, and ≥1601 g.

### 2.2. Statistical Analysis

Phenotypic data collected (body weight of piglets) were treated as a longitudinal trait in the statistical analysis. In addition, repeated measurements (body weights) were treated as level one and observation units (piglets) as level two in the hierarchically structured data. No multiplicative and corrective methods were applied to the original data. Descriptive statistics were obtained by procedures FREQ and MEANS of SAS 9.4 software [16].

Several statistical models with repeated measures were tested before choosing the final model. The MIXED procedure from SAS 9.4 [16] was used for the comparison of the tested statistical models. The criteria for the final selection of the statistical model were based on the Akaike information criterion (AIC), the Schwarz Bayesian information criterion (BIC), and the amount of the residual variance. The random regression coefficient model (RRM), where piglet age was fitted as one quadratic regression per piglet in the random part of the model, was selected as the final model for inferential statistical analysis:y_ijklm_ = BW_i_ + LS_k_ + P_l_ + S_m_ + β_0_ + β_1_ (*t*) + β(t^2^) b_0_ + b_l_(t) + b_2_ (t^2^) +ε_ijkl_
where:y_ijklm_ is live weight;BW_j_ is the effect of the birth weight class; 1–5;LS_k_ is the effect of the litter size class, 1–5;P_l_ is the effect of parity;S_m_ is the effect of the sex; barrows, female;t_ijklm_ is the time of the measuring observation y_ijklm;_β_0_ is the overall intercept for the jklm levels of BWj, LSk, P_l_, S_m;_β_1_ is the overall slope (linear term) for the jklm levels of BW_j_, LS_k_, P_l_, S_m;_β_2_ is the overall slope (quadratic term) for the jklm levels of BW_j_, LS_k_, P_l_, S_m;_b_0_ is the intercept deviation for subject i;b_1i_ is the slope deviation (linear) for subject i;b_2i_ is the slope deviation (quadratic) for subject i;ε is an independent error term distributed normally with mean 0 and variance *σ*^2^.

The variance–covariance matrix of the regression parameters was as follows:$$\left[\begin{array}{ccc} \sigma^{2}{}_{bo} & \sigma^{2}{}_{bob1} & \sigma^{2}{}_{bob1b2}\\ \sigma^{2}{}_{bob1} & \sigma^{2}{}_{b1} & \sigma^{2}{}_{b1b2}\\ \sigma^{2}{}_{bοb1b2} & \sigma^{2}{}_{b1b2} & \sigma^{2}{}_{b2} \end{array}\right]$$

A very important feature of this model is the covariance function, which describes the variance–covariance structure between repeated measurements in time, which in turn allows the estimation of the within-subject covariance between measurements at any two points in time of age, denoted as t_j_ and t_j′_, as well as the between-subject variance at age t_j_.

Graphic visualisation was prepared using the ggplot2 package [17] from R statistical software [18].

## 3. Results

### 3.1. Descriptive Statistics

Castrates and females were almost equally represented in the dataset (Table 3), which is in accordance with the theoretical expectation regarding the sex distribution.

The average NBA was 15.71, followed by a high dispersion of data around the mean with extreme minimum and maximum observations (Table 4), indicating a large number of large litters: more than 66% of the litters were larger than 16 piglets (Table 2). The high variability was also observed in the birth weight of piglets and in the final weight at the end of the experiment. The high variability of the birth weight reflects the problem that can result with the impaired viability of piglets with low birth weights [2,19].

### 3.2. Random Regression Coefficient Analysis

#### Covariance Structure

The random regression coefficient model applied to growth data of the pigs is a method of choice for the analysis of this trait since it can be considered to be a longitudinal trait. In addition, an important advantage of the model used was that the time between measuring needs to be equal and the number of observations can be different [7]. Moreover, after analysing the covariance structure between measurements (Table 5), the assumption for the use of the model that accounts for heterogenous covariances between measurements has been fulfilled.

The result is a single regression line for every piglet in the dataset (Figure 1). The main purpose of using a random regression coefficient model was to assess the most important factors that can affect the early growth of piglets.

From all sources of variation (Table 6) included in the statistical model for live weight, birth weight had the most influential effect, while sex had no statistically significant effect on the final body weight.

A clear pattern of the slower growth of the piglets with lower birth weights can be observed (Figure 2). Piglets from class 1, with the lowest birth weights, had the lowest final weights at the end of the experiment. On the other hand, piglets with high birth weights had the highest body weights at the end of the experiment.

Litter size class showed a strong influential effect on the body weight at the end of the nursing period at the age of 85 days (Figure 3). Estimated growth curves from the model showed that piglets from the LS class 1, i.e., piglets from the smallest litters had significantly higher body weights at the end of the experiment, while piglets from the LS class 5 achieved the lowest values for body weight at the age of 85 days.

Parity had a significant effect on the final body weight of the piglets (Figure 4). The piglets born in the first parity had the highest body weights at the end of the nursery period, while piglets from the second parity had the lowest body weights at the end of the test (Figure 3).

No significant differences between the sexes of the piglets in the analysed dataset were found.

## 4. Discussion

The covariance structure between measurements justified the application of a random regression coefficient model on the collected data. One of the most important features of random regression coefficient models is the ability to use it as an unstructured model and take different variances into account for each period and different covariances between periods [7]. This might be particularly important when considering the litter size of highly prolific sows since there is a big variability in the number of liveborn piglets between parities. Moreover, the large variability in their birth weights collectively makes the growth traits of the piglets one of the most important factors in efficient fattening. Thus, the use of a method that can describe the individual growth of piglets is a method of choice for the analysis of the growth data. In large litters, within-litter variation in birth weight might significantly affect future growth. Thus, the application of a random regression coefficient model allows for the analysis of the growth of every individual, which can be useful for the piglets with variable birth weights. Due to the heterogeneous covariance structure between measurements, classical approaches, such as classical fixed models and random linear models, are not sufficient to describe the growth pattern of piglets since the data structure does not fulfil the assumptions of the homogeneous variance and balanced data [20]. The literature describing the use of models that account for heterogeneous variance between measurements in a random regression coefficient model in the growth of piglets, especially for the analysis of non-genetic factors, is scarce.

The application of the random regression coefficient model confirmed a slower growth in piglets with lower birth weights. The ability of a random regression coefficient model to describe individual growth patterns of piglets can be helpful for breeders in forming groups and adopting feeding and regimes and in planning staying capacities in the farm. The results correspond to earlier studies by Berard et al. [21], who found the significant effect of birth weight groups on myogenesis in piglets. Gondred et al. [22] found a significant association between low birth weight and reduced average daily gain during suckling and the postweaning period. A similar conclusion was brought by Vaclakova et al. [23], who also found an association between low birth weights and impaired growth rates during postweaning and the fattening period. It is commonly recognised that low birth weight in piglets correlates with decreased survival and lower postnatal growth rates. In the majority of low birth weight piglets, low numbers of muscle fibres differentiate during prenatal myogenesis, for genetic or maternal reasons, and those low birth weight piglets with reduced fibre numbers are unable to exhibit postnatal catch-up growth [24]. The strong effect of birth weight on the growth of piglets is relevant to discussions regarding upper limits in the selection for litter sizes and its final effect in managing sow reproduction. Ocepek et al. [25] discussed the sustainability of further artificial selection for litter size in pigs, suggesting that further genetic improvement for litter size might be unsustainable because increments in the number of piglets weaned have increasing costs, such as sibling competition, mortality, and uneven growth, which compromises piglet welfare and fitness.

Riddersholm et al. [2] found that the main critical risk factor for low birth weights observed was the litter size. According to Ocepek et al. [25], piglets from large litters had significantly lower and more variable body weights at weaning. Such a variation might lead to non-uniform piglets in the nursing and fattening period. The negative effects of high litter sizes could be connected to a lower uterine blood flow per foetus when litter size increases [26] and a lack of space in the uterus as a consequence of overcrowded uterus horns, where embryos that were implanted first also prevent the development of additional embryos [5]. Moreover, larger piglets in the litter will have more access to teats or milk replacer during the suckling period, resulting with better growth than in smaller piglets in the litter [27]. On the other hand, Božičković et al. [28] found no statistically significant effect of the litter size on the final weight of piglets at the end of the fattening period, although the birth weight of piglets was higher in animals from small litters.

Milligan et al. [3] found a significant effect of parity on weights at weaning and they reported a higher birth weight of piglets farrowed in the first parity compared to subsequent parities. On the contrary, Carney et al. [29] found that growth performance in the nursery may be affected by dam parity, where the results of the study suggested that the progeny born in higher parities have increased body weight and growth performance during the nursery phase of production compared to piglets born in the first parity. The significant effect of parity on the birth weight of piglets and consequently on the growth of piglets was reported by Amatucci et al. [30], who found that a litter's average daily gain and final weights were higher in multiparous sows than in gilts, probably due to the differences in the colostrum composition within different parities. No effect of parity on the weaning weight was found in the study of Akdag et al. [31]. These results suggest that the piglets in the analysis were not affected by differences in sex when they were in the same environmental conditions such as microclimate and rearing density. Additionally, the reduction of sex difference on the growth rate is considered in the new paternal line because results by Cisneros et al. [32] showed that the sex difference could vary with the genotype. This is in accordance with studies by Škorjanc et al. [33], Bocian et al. [34], and Lee et al. [35]. However, straightforward comparison between different studies might be affected by different experimental conditions and methodologies applied throughout the experiments. According to Kielly et al. [36] castration at 3 days of age can temporarily reduce weight gain. Since castration occurred at the third day of life, the impact of early castration on growth has been accounted for.

## 5. Conclusions

The random coefficient model applied to growth data for piglets from large litters showed the ability to describe the nature of the growth of piglets from birth to the 85th day of life. The birth weight of piglets was the most influential factor on the final weight at the end of the nursing period. On the other hand, the sex of the piglets showed no significant influence on the final weight. The results obtained by using the random regression coefficient model in our study might be encouraging for further applications of this approach in the growth analysis of pigs. Moreover, the practical contribution of the paper is deeper insight into the growth patterns of piglets from highly prolific sows in farm conditions with an emphasis on the need to control the variability of birth weights of large litters. Although pigs are raised in groups, identifying individuals with impaired or good growth ability could help breeders to form groups, adjust feeding and regimes, and adjust planning staying capacities in the farm. This is particularly important for large litters of highly productive sows, where the initial weight at birth can affect future growth. As one of the biggest challenges in managing large litters is to avoid non-heterogeneous body weights of piglets at similar ages, this could be a suitable tool to help achieve this goal.

## Figures and Tables

**Figure 1 animals-13-02888-f001:**
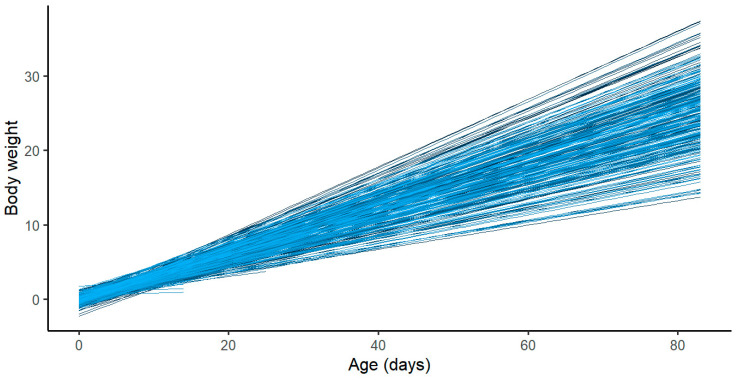
Estimated growth curves for piglets from birth to 85 days of age.

**Figure 2 animals-13-02888-f002:**
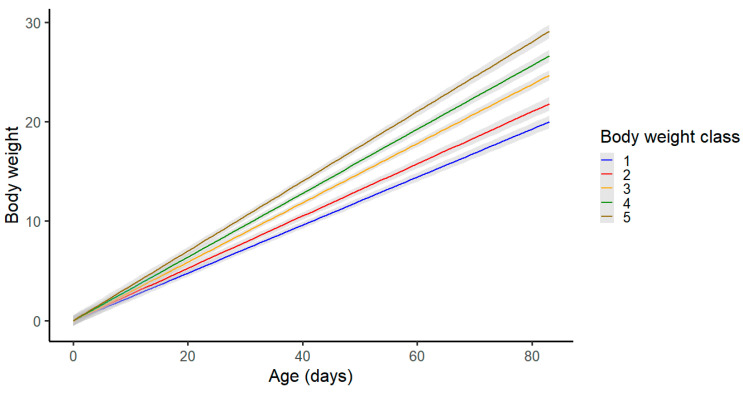
Effect of the birth weight class on body weight at 85 days of age.

**Figure 3 animals-13-02888-f003:**
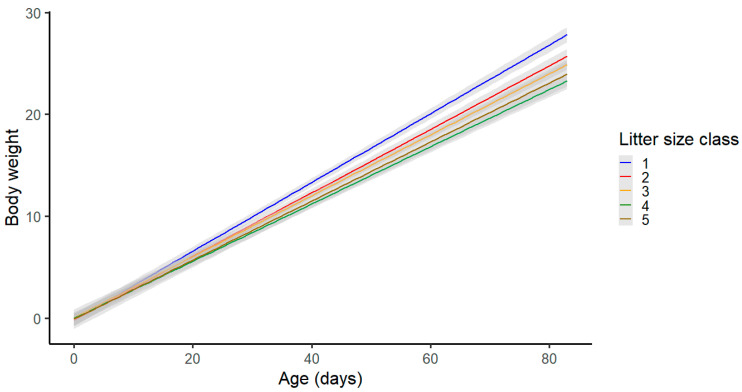
Effect of the litter size class on body weight at 85 days of age.

**Figure 4 animals-13-02888-f004:**
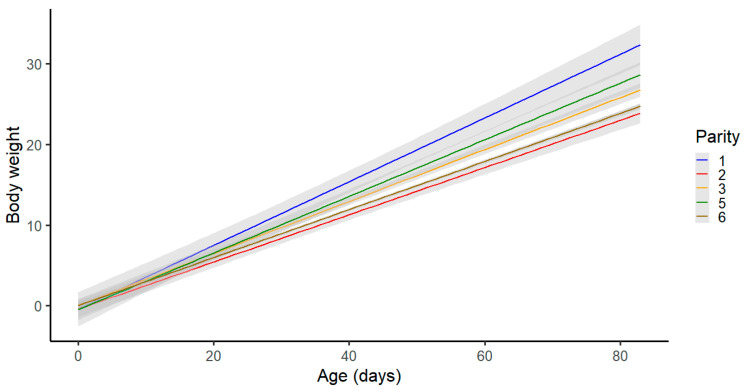
Effect of parity on body weight at 85 days of age.

**Table 1 animals-13-02888-t001:** Detailed chemical composition of mixtures for piglets.

	Pre-Starter	Starter	Grover
Protein, %	17.56	16.5	16.70
Fibre, %	3.20	3.10	4.04
Lipid, %	4.60	5.20	4.55
Ash, %	6.00	7.00	6.50
Lysin, %	1.4	1.44	1.31
Methionine, %	0.37	0.44	0.44
Calcium, %	0.60	0.70	0.68
Phosphorus, %	5.57	0.64	0.49
Sodium, %	0.29	0.21	0.23

**Table 2 animals-13-02888-t002:** Distribution of piglets over litter size and birth weight groups.

Number of Piglets Born Alive	Class	Number of Piglets	Proportion	Birth Weight (g)	Class	Number of Piglets	Proportion
10–12	1	57	15.83	≤1000	1	57	15.83
13–15	2	65	18.06	1001–1200	2	52	14.44
16–18	3	163	45.28	1201–1400	3	103	28.61
19–21	4	56	15.56	1401–1600	4	69	19.17
>21	5	19	5.28	≥1601	5	79	21.94

**Table 3 animals-13-02888-t003:** Frequency of male and female piglets and the total number of piglets in the dataset.

Number of litters	25
Number of male piglets/castrates	194
Number of female piglets	166
Total number of piglets	360

**Table 4 animals-13-02888-t004:** Descriptive statistics for the number of piglets born alive, stillborn, birth and final weight.

Parameter	Mean	SD	CV	Min	Max
NBA	15.71	3.10	19.71	6	21
Stillborn	2.18	2.65	121.40	0	9
Birth weight, kg	1.33	0.33	25.17	0.40	2.22
Final weight (85 d)	27.96	5.30	18.96	15.40	42.00

**Table 5 animals-13-02888-t005:** Covariance matrix between measurements.

	1	2	3	4	5
1	1.0000	0.4033	−0.2706	0.2732	−0.4601
2		1.0000	0.7114	0.1894	0.1003
3			1.0000	−0.1079	0.3426
4				1.0000	0.5670
5					1.0000

**Table 6 animals-13-02888-t006:** The significance of the main effects in the model for live weight at 85 days.

Source	DF	F Value	*p* Value
Birth weight	1	651.53	<0.0001
Litter size class	4	7.69	<0.0001
Parity	4	9.42	<0.0001
Sex	1	0.02	0.8875

## Data Availability

Data are available upon request to the authors.

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
