# Peer review of "Analysis of Early Growth of Piglets from Hyperprolific Sows Using Random Regression Coefficient"

_animals, 2023, doi:10.3390/ani13182888_

Round 1

Reviewer 1 Report (Previous Reviewer 1)

The manuscript has been significantly improved and now warrants publication in animals.

Author Response

Thank you for your comment. 

Reviewer 2 Report (Previous Reviewer 2)

I again read the manuscript carefully and I still have the point of view, beside low scientific impact and ignoring the trend of global pig breeding goals, that this study should be rejected. 

Modelling is done correctly - a little improvement could be made, and also the paper is well written - however authors totally ignore the effect of a too early induced birth - which is well know, that too early induced birth affect piglet vitality at birth, increases the number of dead born piglets, and effects viability and development of piglets the entire suckling period. 

I stated this concern in the last review - no comment on my concern was made by the authors. 

I totally understand the statistical aim of their work, nevertheless I cannot overlook the study design's flaw of unbiased data, which does not correspond to standards of good scientific practices and animal welfare. 

Author Response

Please, see attachment. 

This manuscript is a resubmission of an earlier submission. The following is a list of the peer review reports and author responses from that submission.

Round 1

Reviewer 1 Report

1. Reference labeling needs to be standardized, such as line 64,[6-8]

2. It is recommended to compare the random regression coefficient model with other methods to demonstrate the advantages of using the method.

3. It is recommended to write the results separately from the discussion section, and the discussion section should be further strengthened.

4. The author's contribution needs to be clearly described

Reviewer 2 Report

I reject the paper due to study design and animal welfare concerns, besides low to moderate statistics and little scientific input in this research area.

Overall statement: it is well known that birth weight and live weight over time is highly correlated.

Study design and animal welfare:

Croatia is part of EU animal welfare legislation, therefore it is critical do tail dock and clip teeth without ANY stated reason in this study. In addition treating sows with d-cloprostenol on day 112 can still be considered as critical. Such a treatment before day 114 can have bad effects on sows milk performance and piglet vitality which influences gain in live weight. 

It is not clear to me how dead born piglets are defined and assessed and if they were weighted.

Statistics: Setting up classes: should have been set up on total number of born piglets.  Model terms are not well discribed. 

Have authors tried to fit any splines instead of linear and quadratic terms?

Moderate English 
